# Photonic Differential Privacy with Direct Feedback Alignment

**Ruben Ohana**[*1,3] , **Hamlet J. Medina Ruiz**[*2], **Julien Launay**[*1,3], **Alessandro Cappelli**[1],
**Iacopo Poli**[1], **Liva Ralaivola**[2], **Alain Rakotomamonjy**[2]

[1]LightOn, Paris, France
[2]Criteo AI Lab, Paris, France
[3]LPENS, École Normale Supérieure, Paris, France

## Abstract

Optical Processing Units (OPUs) – low-power photonic chips dedicated to large scale random projections – have been used in previous work to train deep neural networks using Direct Feedback Alignment (DFA), an effective alternative to backpropagation. Here, we demonstrate how to leverage the intrinsic noise of optical random projections to build a differentially private DFA mechanism, making OPUs a solution of choice to provide a *private-by-design* training. We provide a theoretical analysis of our adaptive privacy mechanism, carefully measuring how the noise of optical random projections propagates in the process and gives rise to provable Differential Privacy. Finally, we conduct experiments demonstrating the ability of our learning procedure to achieve solid end-task performance.

## 1 Introduction

The widespread use of machine learning models has created concern about their release in the wild when trained on sensitive data such as health records or queries in data bases [7, 15]. Such concern has motivated a abundant line of research around privacy-preserving training of models. A popular technique to guarantee privacy is *differential privacy* (DP), that works by injecting noise in an deterministic algorithm, making the contribution of a single data-point hardly distinguishable from the added noise. Therefore it is impossible to infer information on individuals from the aggregate.

While there are alternative methods to ensure privacy, such as knowledge distillation (e.g. PATE [25]), a simple and effective strategy is to use perturbed and quenched Stochastic Gradient Descent (SGD) [1]: the gradients are clipped before being aggregated and then perturbed by some additive noise, finally they are used to update the parameters. The DP property comes at a cost of decreased utility. These biased and perturbed gradients provide a noisy estimate of the update direction and decrease utility, i.e. end-task performance.

We revisit this strategy and develop a private-by-design learning algorithm, inspired by the implementation of an alternative training algorithm, Direct Feedback Alignment [23], on Optical Processing Units [20], photonic co-processors dedicated to large scale random projections. The analog nature of the photonic co-processor implies the presence of noise, and while this is usually minimized, in this case we propose to leverage it, and tame it to fulfill our needs, *i.e* to control the level of privacy of the learning process. The main sources of noise in Optical Processing Units can be modeled as additive Poisson noise on the output signal, and approach a Gaussian distribution in the operating regime of the device. In particular, these sources can be modulated through temperature control, in order to attain a desired privacy level.

Finally, we test our algorithm using the photonic hardware demonstrating solid performance on the goal metrics. To summarize, our setting consists in OPUs performing the multiplication by a fixed random matrix, with a different realization of additive noise for every random projection.

---

*Equal contribution. Corresponding authors: `ruben@lighton.ai` and `hj.medinaruiz@criteo.com`

35th Conference on Neural Information Processing Systems (NeurIPS 2021).

## 1.1 Related work

The amount of noise needed to guarantee differential privacy was first formalized in [9]. Later, a training algorithm that satisfied Renyi Differential Privacy was proposed in [1]. This sparked a line of research in differential privacy for deep learning, investigating different architecture and clipping or noise injection mechanisms [2, 3]. The majority of these works though rely on backpropagation. An original take was offered in [19], that evaluated the privacy performance of Direct Feedback Alignment (DFA) [23], an alternative to backpropagation. While Lee et al. [19] basically extend the gradient clipping/Gaussian mechanism approach to DFA, our work, while applied to the same DFA setting, is motivated by a photonic implementation that naturally induces noise that we exploit for differential privacy. As such, we provide a new DP framework together with its theoretical analysis.

## 1.2 Motivations and contributions

We propose a hardware-based approach to Differential Privacy (DP), centered around a photonic co-processor, the OPU. We use it to perform optical random projections for a differentially private DFA training algorithm, leveraging noise intrinsic to the hardware to achieve *privacy-by-design*. This is a significant departure from the classical view that such analog noise should be minimized, instead leveraging it as a key feature of our approach. Our mechanism can be formalized through the following (simplified) learning rule at layer $\ell$:

$$\delta\mathbf{W}^\ell = -\eta[(\underbrace{\mathbf{B}^{\ell+1}\mathbf{e}}_{\text{scaled DFA learning signal}} + \overbrace{\boldsymbol{g}}^{\text{Gaussian hardware noise}}) \odot \phi'_\ell(\mathbf{z}^\ell)](\underbrace{\mathbf{h}^{\ell-1}}_{\text{neuron-wise clipped activations}})^\top \tag{1}$$

**Photonic-inspired and tested.** The OPU is used as both inspiration and an actual platform for our experiments. We demonstrate theoretically that the noise induced by the analog co-processor makes the algorithm private by design, and we perform experiments on a real photonic co-processor to show we achieve end-task performance competitive with DFA on GPU.

**Differential Privacy beyond backpropagation.** We extend previous work [19] on DP with DFA both theoretically and empirically. We provide new theoretical elements for noise injected on the DFA learning signal, a setting closer to our hardware implementation.

**Theoretical contribution.** Previous works on DP and DFA [19] proposes a straightforward extension of the DP-SGD paradigm to direct feedback alignment. In our work, by adding noise directly to the random projection in DFA, we study a different Gaussian mechanism [22] with a covariance matrix depending on the values of the activations of the network. Therefore the theoretical analysis is more challenging than in [19]. We succeed to upper bound the Rényi Divergence of our mechanism and deduce the $(\epsilon, \delta)$-DP parameters of our setup.

## 2 Background

Formally the problem we study the following: the analysis of the built-in Differential Privacy thanks to the combination of DFA and OPUs to train deep architectures. Before proceeding, we recall a minimal set of principles of DFA and Differential Privacy.

From here on $\{\boldsymbol{x}_i\}_{i=1}^N$ are the training points belonging to $\mathbb{R}^d$, $\{y_i\}_{i=1}^N$ the target labels belonging to $\mathbb{R}$. The aim of training a neural network is to find a function $f : \mathbb{R}^d \to \mathbb{R}$ that minimizes the *true $\mathcal{L}$-risk* $\mathbb{E}_{XY\sim D}\mathcal{L}(f(X), Y)$, where $\mathcal{L} : \mathbb{R} \times \mathbb{R} \to \mathbb{R}$ is a loss function and $D$ a fixed (and unknown) distribution over data and labels (and the $(\boldsymbol{x}_i, y_i)$ are independent realizations of $X, Y$), and to achieve that, the *empirical* risk $\frac{1}{N}\sum_{i=1}^N \mathcal{L}(f(\boldsymbol{x}_i), y_i)$ is minimized.

### 2.1 Learning with Direct Feedback Alignment (DFA)

DFA is a biologically inspired alternative to backpropagation with an asymmetric backward pass. For ease of notation, we introduce it for fully connected networks but it generalizes to convolutional networks, transformers and other architectures [17]. It has been theoretically studied in [21, 27]. Note that in the following, we incorporate the bias terms in the weight matrices.

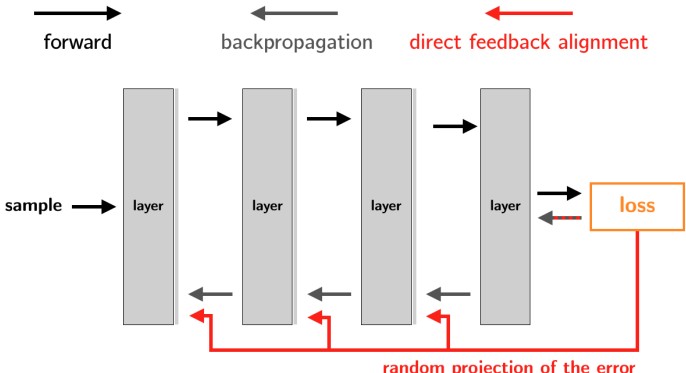

Figure 1: Schematic comparison of backpropagation and direct feedback alignment. The two approaches differ in how the loss impacts each layer of the model. While in backpropagation, the loss is propagated sequentially backwards, in DFA, it directly acts on each layer after random projection.

**Forward pass.** In a model with $L$ layers of neurons, $\ell \in \{1, \ldots, L\}$ is the index of the $\ell$-th layer, $\mathbf{W}^\ell \in \mathbb{R}^{n_\ell \times n_{\ell-1}}$ the weight matrix between layers $\ell - 1$ and $\ell$, $\phi_\ell$ the activation function of the neurons, and $\boldsymbol{h}_\ell$ their activations. The forward pass for a pair $(\boldsymbol{x}, y)$ writes as:

$$\forall \ell \in \{1, \ldots, L\} : \mathbf{z}_\ell = \mathbf{W}^\ell \mathbf{h}^{\ell-1}, \mathbf{h}^\ell = \phi_\ell(\mathbf{z}^\ell), \tag{2}$$

where $\mathbf{h}^0 \doteq \boldsymbol{x}$ and $\hat{\mathbf{y}} \doteq \mathbf{h}^L = \phi(\mathbf{z}^L)$ is the predicted output.

**Backpropagation update.** With backpropagation [28], leaving aside the specifics of the optimizer used (learning rate, etc.), the weight updates are computed using the chain-rule of derivatives :

$$\delta \mathbf{W}^\ell = -\frac{\partial \mathcal{L}}{\partial \mathbf{W}^\ell} = -[((\mathbf{W}^{\ell+1})^\top \delta \mathbf{z}^{\ell+1}) \odot \phi_\ell'(\mathbf{z}^\ell)](\mathbf{h}^{\ell-1})^\top, \delta \mathbf{z}^\ell = \frac{\partial \mathcal{L}}{\partial \mathbf{z}^\ell}, \tag{3}$$

where $\phi_\ell'$ is the derivative of $\phi_\ell$, $\odot$ is the Hadamard product, and $\mathcal{L}(\hat{\mathbf{y}}, \mathbf{y})$ is the prediction loss.

**DFA update.** DFA replaces the gradient signal $(\mathbf{W}^{\ell+1})^\top \delta \mathbf{z}^{\ell+1}$ with a random projection of the derivative of the loss with respect to the pre-activations $\delta \boldsymbol{z}^L$ of the last layer. For losses $\mathcal{L}$ commonly used in classification and regression, such as the squared loss or the cross-entropy loss, this will amount to a random projection of the error $\mathbf{e} \propto \hat{\mathbf{y}} - \mathbf{y}$. With a fixed random matrix $\mathbf{B}^{\ell+1}$ of appropriate shape drawn at initialization of the learning process, the parameter update of DFA is:

$$\delta \mathbf{W}^\ell = -[(\mathbf{B}^{\ell+1}\mathbf{e}) \odot \phi_\ell'(\mathbf{z}^\ell)](\mathbf{h}^{\ell-1})^\top, \mathbf{e} = \frac{\partial \mathcal{L}}{\partial \boldsymbol{z}^L} \tag{4}$$

**Backpropagation vs DFA training.** Learning using backpropagation consists in iteratively applying the forward pass (2) on batches of training examples and then applying backpropagation updates (3). Training with DFA consists in replacing the backpropagation updates by DFA ones (4). An interesting feature of DFA is the parallelization of the training step, where all the random projections of the error can be done at the same time, as shown in Figure 1.

## 2.2 Optical Processing Units

An Optical Processing Unit (OPU)[2] is a co-processor that multiplies an input vector $\boldsymbol{x} \in \mathbb{R}^d$ by a fixed random matrix $\mathbf{B} \in \mathbb{R}^{D \times d}$, harnessing the physical phenomenon of light scattering through a diffusive medium [20]. The operation performed is:

$$\boldsymbol{p} = \mathbf{B}\boldsymbol{x} \tag{5}$$

If the coefficients of $\mathbf{B}$ are unknown, they are guaranteed to be (independently) distributed according to a Gaussian distribution [20, 24]. An additional interesting characteristics of the OPU is its low energy consumption compared to GPUs for high-dimensional matrix multiplication [24].

---

[2]Accessible through LightOn Cloud: `https://cloud.lighton.ai`.

A central feature we will rely on is that the measurement of the random projection (5) may be corrupted by an additive zero-mean Gaussian random vector $\boldsymbol{g}$, so as for an OPU to provide access to $\boldsymbol{p} = \mathbf{B}\boldsymbol{x} + \boldsymbol{g}$. If $\boldsymbol{g}$ is usually negligible, its variance can be modulated by controlling the physical environment around the OPU. We take advantage of this feature to enforce differential privacy. In the current versions of the OPUs, however, modulating the analog noise at will is not easy, so we will *simulate* the noise numerically in the experiments.

## 2.3 Differential Privacy (DP)

Differential Privacy [9, 10] sets a framework to analyze the privacy guarantees of algorithms. It rests on the following core definitions.

**Definition 2.1** (Neighboring datasets). Let $\{\mathcal{X}^j\}_{j=1}^N$ (e.g. $\mathcal{X}^j = \mathbb{R}^d$) be a *domain* and $\mathcal{D} \doteq \cup_{j=1}^N \mathcal{X}^j$. $D, D' \in \mathcal{D}$ are *neighboring datasets* if they differ from one element. This is denoted by $\mathcal{D} \sim \mathcal{D}'$.

**Definition 2.2** (($\varepsilon, \delta$)-differential privacy [10]). Let $\varepsilon, \delta > 0$. Let $\mathcal{A} : \mathcal{D} \rightarrow \text{Range}\mathcal{A}$ be a *randomized* algorithm, where $\text{Range}\mathcal{A}$ is the range of $\mathcal{D}$ through $\mathcal{A}$. $\mathcal{A}$ is ($\varepsilon, \delta$)-differentially private, or ($\varepsilon, \delta$)-DP, if for all neighboring datasets $D, D' \in \mathcal{D}$ and for all sets $\mathcal{O} \in \text{Im } \mathcal{A}$, the following inequality holds:

$$\mathbb{P}[\mathcal{A}(D) \in \mathcal{O}] \leq e^\varepsilon \mathbb{P}[\mathcal{A}(D') \in \mathcal{O}] + \delta$$

where the probability relates to the randomness of $\mathcal{A}$.

Mironov [22] proposed an alternative notion of differential privacy based on Rényi $\alpha$-divergences and established a connection between their definition and the ($\varepsilon, \delta$)-differential privacy of Definition 2.2. Rényi-based Differential Privacy is captured by the following:

**Definition 2.3** (Rényi $\alpha$-divergence [29]). For two probability distributions $P$ and $Q$ defined over $\mathbb{R}$, the Rényi divergence of order $\alpha > 1$ is given by:

$$\mathbb{D}_\alpha \left( P \| Q \right) \doteq \frac{1}{\alpha - 1} \log \mathbb{E}_{x \sim Q} \left( \frac{P(x)}{Q(x)} \right)^\alpha \tag{6}$$

**Definition 2.4** (($\alpha, \varepsilon$)-Rényi differential privacy [22]). Let $\varepsilon > 0$ and $\alpha > 1$. A randomized algorithm $\mathcal{A}$ is ($\alpha, \varepsilon$)-Rényi differential private or ($\alpha, \varepsilon$)-RDP, if for any neighboring datasets $D, D' \in \mathcal{D}$,

$$\mathbb{D}_\alpha \left( \mathcal{A}(D) \| \mathcal{A}(D') \right) \leq \varepsilon.$$

**Theorem 1** (Composition of RDP mechanisms [22]). *Let $\{M_i\}_{i=1}^k$ be a set of mechanisms, each satisfying $(\alpha, \epsilon_i)$-RDP. Then their combination is $(\alpha, \sum_i \epsilon_i)$-RDP.*

Going from RDP to the Differential Privacy of Definition 2.2 is made possible by the following theorem (see also [4, 6, 30]).

**Theorem 2** (Converting RDP parameters to DP parameters [22]). *An $(\alpha, \varepsilon)$-RDP mechanism is $\left( \varepsilon + \frac{\log 1/\delta}{\alpha - 1}, \delta \right)$-DP for all $\delta \in (0, 1)$.*

For the theoretical analysis, we will need the following proposition, specific to the case of multivariate Gaussian distributions, to obtain bounds on the Rényi divergence.

**Proposition 1** (Rényi divergence for two multivariate Gaussian distributions [26]). *The Rényi divergence for two multivariate Gaussian distributions with means $\boldsymbol{\mu}_1, \boldsymbol{\mu}_2$ and respective covariances $\boldsymbol{\Sigma}_1, \boldsymbol{\Sigma}_2$ is given by:*

$$\mathbb{D}_\alpha(\mathcal{N}(\boldsymbol{\mu}_1, \boldsymbol{\Sigma}_1) \| \mathcal{N}(\boldsymbol{\mu}_2, \boldsymbol{\Sigma}_2)) = \frac{\alpha}{2}(\boldsymbol{\mu}_1 - \boldsymbol{\mu}_2)^\top (\alpha \boldsymbol{\Sigma}_2 + (1 - \alpha)\boldsymbol{\Sigma}_1)^{-1}(\boldsymbol{\mu}_1 - \boldsymbol{\mu}_2)$$
$$- \frac{1}{2(\alpha - 1)} \log \left[ \frac{\det(\alpha \boldsymbol{\Sigma}_2 + (1 - \alpha)\boldsymbol{\Sigma}_1)}{(\det \boldsymbol{\Sigma}_1)^{1-\alpha}(\det \boldsymbol{\Sigma}_2)^\alpha} \right] \tag{7}$$

*with $\det(\boldsymbol{\Sigma})$ the determinant of the matrix $\boldsymbol{\Sigma}$. Note that $(\alpha \boldsymbol{\Sigma}_2 + (1 - \alpha)\boldsymbol{\Sigma}_1)^{-1}$ must be definite-positive[3], otherwise the Rényi divergence is not defined and is equal to $+\infty$.*

---

[3]Note that here $\alpha > 1$ and the combination $\alpha \boldsymbol{\Sigma}_2 + (1 - \alpha)\boldsymbol{\Sigma}_1$ is *not* a convex combination; extra-case must be given to ensure that the resulting matrix is positive

**Remark 2.1.** A standard method to generate an (R)DP algorithm from a deterministic function $\boldsymbol{f} : \mathcal{X} \to \mathbb{R}^d$ is the *Gaussian mechanism* $\mathcal{M}_\sigma$ acting as $\mathcal{M}_\sigma \boldsymbol{f}(\cdot) = \boldsymbol{f}(\cdot) + \boldsymbol{v}$ where $\boldsymbol{v} \sim \mathcal{N}(0, \sigma^2 \mathbf{I}_d)$. If $\boldsymbol{f}$ has $\Delta_{\boldsymbol{f}}$- (or $\ell_2$-) *sensitivity*

$$\Delta_{\boldsymbol{f}} \doteq \max_{D \sim D'} \|\boldsymbol{f}(D) - \boldsymbol{f}(D')\|_2,$$

then $\mathcal{M}_\sigma$ is $\left(\alpha, \frac{\alpha \Delta_{\boldsymbol{f}}^2}{2\sigma^2}\right)$-RDP.

# 3 Photonic Differential Privacy

This section explains how to use photonic devices to perform DFA and a theoretical analysis showing how this combination entails *photonic differential privacy*.

## 3.1 Clipping parameters

As usual in Differential Privacy, we will need to clip layers of the network during the backward pass. Given a vector $\boldsymbol{v} \in \mathbb{R}^d$ and positive constants $c, s, \nu$, we define:

- $\mathrm{clip}_c(\boldsymbol{v}) \doteq (\mathrm{sign}(v_1) \cdot \min(c, |v_1|), \ldots, \mathrm{sign}(v_d) \cdot \min(c, |v_d|))^\top$
- $\mathrm{scale}_s(\boldsymbol{v}) \doteq \min(s, \|\boldsymbol{v}\|_2) \frac{\boldsymbol{v}}{\|\boldsymbol{v}\|_2}$
- $\mathrm{offset}_\nu(\boldsymbol{v}) \doteq (v_1 + \nu, \ldots, v_d + \nu)^\top$

The weight update with clipping to be considered in place of (4) is given by

$$\delta \mathbf{W}^\ell = -\frac{1}{m} \sum_{i=1}^m (\mathrm{scale}_{s_\ell}(\mathbf{B}^\ell \boldsymbol{e}_i) + \boldsymbol{g}_i) \odot \phi'(\boldsymbol{z}_i^\ell)) \mathrm{clip}_{c_\ell}(\mathrm{offset}_{\nu_\ell}(\boldsymbol{h}_i^\ell))^\top \tag{8}$$

For each layer $\ell$, we set the $s_\ell$, $c_\ell$ and $\nu_\ell$ parameters as follows:

$$c_\ell \doteq \frac{\tau_h^{max}}{\sqrt{n_\ell}} \qquad \nu_\ell \doteq \frac{\tau_h^{min}}{\sqrt{n_\ell}} \qquad s_\ell \doteq \tau_B \tag{9}$$

These choices ensure that:

- $\tau_h^{min} \leq \|\mathrm{clip}_{c_\ell}(\mathrm{offset}_{\nu_\ell}(\boldsymbol{h}_i^\ell))\|_2 \leq \tau_h^{max}$
- $\|\mathrm{scale}_{s_\ell}(\mathbf{B}^\ell \boldsymbol{e}_i)\|_2 \leq \tau_B$
- Moreover, we require the derivatives of the each activation function $\phi_\ell$ are lower and upper bounded by constants i.e. $\gamma_\ell^{min} \leq |\phi_\ell'(t)| \leq \gamma_\ell^{max}$ for all scalars $t$. This is a reasonable assumption for activation functions such as sigmoid, tanh, ReLU...

In the following, the quantities should all be considered clipped/scaled/offset as above and, for sake of clarity, we drop the explicit mentions of these operations.

## 3.2 Photonic Direct Feedback Alignment is a natural Gaussian mechanism

**Noise modulation.** Mainly due to the photon shot noise of the measurement process [16], Gaussian noise is naturally added to the random projection of (5). This noise is negligible for machine learning purposes, however it can be modulated through temperature control yielding the following projection:

$$\boldsymbol{p} = \mathbf{B}\boldsymbol{x} + \mathcal{N}(0, \sigma^2 \mathbf{I}_D) \tag{10}$$

where $\mathbf{I}_D$ is the identity matrix in dimension $D$. Note that this noise is *truly random* due to the quantum nature of the photon shot noise. As previously stated, due to experimental constraints, the noise will be simulated in the experiments of Section 4.

Building on that feature, we perform the random projection of DFA of (4) using the OPU. Since this equation is valid for any layer $\ell$ of the network, we allow ourselves, for sake of clarity, to drop the layer indices and study the generic update (the quantities below are all clipped as in Section 3.1):

$$\delta \mathbf{W} = -\frac{1}{m} \sum_{i=1}^m (\mathbf{B}\boldsymbol{e}_i + \boldsymbol{g}_i) \odot \phi'(\boldsymbol{z}_i)) \boldsymbol{h}_i^\top \quad \text{(clipped quantities as in section 3.1)} \tag{11}$$

$$= -\frac{1}{m} \sum_{i=1}^m (\mathbf{B}\boldsymbol{e}_i \odot \phi'(\boldsymbol{z}_i)) \boldsymbol{h}_i^\top + \frac{1}{m} \sum_{i=1}^m (\boldsymbol{g}_i \odot \phi'(\boldsymbol{z}_i)) \boldsymbol{h}_i^\top \tag{12}$$

---
**Algorithm 1** Photonic DFA training
---
**Require:** training set $\mathcal{S} = \{(\boldsymbol{x}_j, y_j)\}_{j=1}^{N}$, $\phi_\ell$ with bounded derivatives, scale parameters $s_\ell$, clipping
    thresholds $\nu_\ell$ and $c_\ell$, stepsize $\eta$, noise scale $\sigma$, minibatch of size $m$, number of iterations $T$
**Ensure:** A performing DP model
  1: **for** $\ell = 1$ to $L$ **do**
  2:     Sample $\mathbf{B}^\ell$                                      ▷ Note: with OPUs, there is no explicit sampling of $B$
  3: **end for**
  4: **for** $T$ iterations **do**
  5:     Create a minibatch $S \subset \{1, \ldots, N\}$ of size $|S| = m$ (sampling without replacement)
  6:     **for** $i \in S$ **do**
  7:         **for** $\ell = 1$ to $L-1$ **do**
  8:             $\boldsymbol{z}_i^\ell = \mathbf{W}^\ell \boldsymbol{h}_i^{\ell-1}$
  9:             $\boldsymbol{h}_i^\ell = \phi_\ell(\boldsymbol{z}_i^\ell)$
10:         **end for**
11:         $\hat{\boldsymbol{y}}_i \leftarrow \phi_\ell(\mathbf{W}^\ell \boldsymbol{h}_i^{\ell-1})$
12:     **end for**
13:     **for** $l = L$ to $1$ **do**
14:         Perform $\mathbf{B}^{\ell+1} \boldsymbol{e}_i$ with the OPU
15:         Independently sample $\boldsymbol{g}_1^\ell, \ldots, \boldsymbol{g}_m^\ell \sim \mathcal{N}(0, \sigma^2 I_D)$
16:         $\mathbf{W}^\ell \leftarrow \mathbf{W}^\ell - \frac{\eta}{m} \sum_{i=1}^{m}((\text{scale}_{s_\ell}(\mathbf{B}^{\ell+1}\boldsymbol{e}_i) + \boldsymbol{g}_i^\ell) \odot \phi_\ell'(\boldsymbol{z}_i^\ell))\text{clip}_{c_\ell}(\text{offset}_{\nu_\ell}(\boldsymbol{h}_i^\ell))^\top$
17:     **end for**
18: **end for**
---

where $\boldsymbol{g}_i \sim \mathcal{N}(0, \sigma^2 I_{n_\ell})$ is the Gaussian noise added during the OPU process. As stated previously, its variance $\sigma^2$ can be modulated to obtain the desired value. The overall training procedure with Photonic DFA (PDFA) is described in Algorithm 3.1.

### 3.3 Theoretical Analysis of our method

In the following, the quantities in the DFA update of the weights are always clipped according to (11) and as before clipping/scale/offset operators are in force but dropped from the text.

To demonstrate that our mechanism is Differentially Private, we will use the following reasoning: the noise being added at the random projection level as in (10), we can decompose the update of the weights as a Gaussian mechanism as in (12). We will compute the covariance matrix of the Gaussian noise, which will depend on the data, which is in striking contrast with the standard Gaussian mechanism [1]. We will then use Proposition 1 to compute the upper bound the Rényi divergence. The Differential Privacy parameters will be obtained using Theorem 2.

In the following, we will consider the Gaussian mechanism applied to the columns of the weight matrix. We consider this case for the following reasons: since our noise matrix has the same realisation of the Gaussian noise (but multiplied by different scalars), it makes sense to consider the Differential Privacy parameters of only columns of the weight matrix and then multiply the Rényi divergence by the number of columns. If our noise was i.i.d. we could have used the theorems from [8] to lower the Rényi divergence. Given the update equation of the weights at layer $l$ in (12), the update of column $k$ of the weight of layer $l$ is the following Gaussian mechanism:

$$\frac{1}{m} \sum_{i=1}^{m}((\mathbf{B}\boldsymbol{e}_i) \odot \phi'(\boldsymbol{z}_i))h_{ik} + \frac{1}{m} \sum_{i=1}^{m}(\boldsymbol{g}_i \odot \phi'(\boldsymbol{z}_i))h_{ik} = \boldsymbol{f}_k(D) + \mathcal{N}(0, \boldsymbol{\Sigma}_k) \qquad (13)$$

where $\boldsymbol{\Sigma}_k = \frac{\sigma^2}{m^2}\mathbf{diag}(\boldsymbol{a}_k)^2$ and $(\boldsymbol{a}_k)_j = \sqrt{\sum_{i=1}^{m}(\phi_{ij}' h_{ik})^2}, \forall j = 1, \ldots, n_{\ell-1}$. Note that these expressions are due to the inner product with $\boldsymbol{h}_i$. In the following, we will focus on column $k$ and we therefore drop the index $k$ in the notation. Using the clipping of the quantities of interest detailed in (8), we can compute some useful bounds bounds on $a_j$:

$$\sqrt{\frac{m}{n_\ell}}\gamma_\ell^{min}\tau_h^{min} \leq a_j \leq \sqrt{\frac{m}{n_\ell}}\gamma_\ell^{max}\tau_h^{max} \qquad (14)$$

**Proposition 2** (Sensitivity of Photonic DFA [19]). *For neighboring datasets $D$ and $D'$ (i.e. differing from only one element), the sensitivity $\Delta_{\boldsymbol{f}}^{\ell}$ of the function $\boldsymbol{f}_k$ described in (12) at layer $l$ is given by:*

$$\Delta_{\boldsymbol{f}}^{\ell} = \sup_{D \sim D'} \|\boldsymbol{f}(D) - \boldsymbol{f}(D')\|_2 \leq \frac{2}{m} \|(\mathbf{B}^{\ell} \boldsymbol{e}_i) \odot \phi_{\ell}'(\boldsymbol{z}_i^{\ell})) h_{ik}^{\ell-1}\|_2 \tag{15}$$

$$\leq \frac{2}{m} \tau_B \gamma_{\ell}^{max} \frac{\tau_h^{max}}{\sqrt{n_{\ell}}} \tag{16}$$

The following proposition is our main theoretical result: we compute the $\epsilon$ parameter of Rényi Differential Privacy.

**Proposition 3** (Photonic Differential Privacy parameters). *Given two probability distributions $P \sim \mathcal{N}(\boldsymbol{f}(D), \boldsymbol{\Sigma})$ and $Q \sim \mathcal{N}(\boldsymbol{f}(D'), \boldsymbol{\Sigma}')$ corresponding to the Gaussian mechanisms depicted in (13) on neighboring datasets $D$ and $D'$, the Rényi divergence of order $\alpha$ between these mechanisms is:*

$$\mathbb{D}_{\alpha}(P\|Q) \leq \frac{2 n_{\ell}.\alpha}{m.\sigma^2} \frac{(\gamma^{max} \tau^{max} \tau_B)^2}{(\gamma_{\ell}^{min} \tau_h^{min})^2} + \frac{n_{\ell}.\alpha}{2(\alpha-1)} \log\left[\frac{m(\gamma_{\ell}^{min} \tau_h^{min})^2}{(m+1)(\gamma_{\ell}^{min} \tau_h^{min})^2 - (\gamma_{\ell}^{max} \tau_h^{max})^2}\right]$$
$$= \varepsilon_{PDFA} \tag{17}$$

*Our mechanism is therefore $(\alpha, T\varepsilon_{PDFA})$-RDP with $T$ the number of training epochs. We can deduce that the mechanism on the weight matrix with $n_{\ell-1}$ columns is $(\alpha, T n_{\ell-1}\varepsilon_{PDFA})$-RDP. Then the mechanism of the whole network composed of $L$ layers is $(\alpha, L T n_{\ell-1}\varepsilon_{PDFA})$-RDP. We can then convert our bound to DP parameters using Theorem 2 to obtain a $(L T n_{\ell-1}\varepsilon_{PDFA} + \frac{\log 1/\delta}{\alpha-1}, \delta)$-DP mechanism for all $\delta \in (0,1)$.*

*Proof.* In the following, the variables with a prime correspond to the ones built upon dataset $D'$. According to (13), the covariance matrices $\boldsymbol{\Sigma}$ and $\boldsymbol{\Sigma}'$ are diagonal and any of their weighted sum is diagonal, as well as their inverse. Moreover, the determinant of a diagonal matrix is the product of its diagonal elements. Using these elements in (7) yield:

$$\mathbb{D}_{\alpha}(P\|Q) = \sum_{j=1}^{n_{\ell}} \left( \frac{\alpha m^2}{2\sigma^2} \frac{(f_j(D) - f_j'(D'))^2}{\alpha\, a_j'^2 + (1-\alpha) a_j^2} - \frac{1}{2(\alpha-1)} \log\left[\frac{(1-\alpha) a_j^2 + \alpha a_j'^2}{a_j^{2(1-\alpha)} a_j'^{2\alpha}}\right]\right)$$

Using the fact that we are studying neighboring datasets, the sums composing $a_j$ and $a_j'$ differ by only one element at element $i = I$. This implies that

$$\alpha\, a_j'^2 + (1-\alpha) a_j^2 = a_j^2 + \alpha[(\tilde{\phi}_{Ij}' \tilde{h}_{Ik})^2 - (\phi_{Ij}' h_{Ik})^2)]$$

where $\tilde{\phi}_{Ij}'$ and $\tilde{h}_{Ik}^2$ are taken on the dataset $D'$. By choosing $D$ and $D'$ such that $[(\tilde{\phi}_{Ij}' \tilde{h}_{Ik})^2 - (\phi_{Ij}' h_{Ik})^2)] \geq 0$ and some rearrangement, we can upper bound the Rényi divergence by:

$$\mathbb{D}_{\alpha}(P\|Q) \leq \frac{\alpha.n_{\ell}.m^2}{2\sigma^2} \frac{\Delta_{\boldsymbol{f}}^2}{(\sqrt{\frac{m}{n_{\ell}}} \gamma_{\ell}^{min} \tau_h^{min})^2} + \frac{\alpha.n_{\ell}}{2(\alpha-1)} \log\left[\frac{m(\gamma_{\ell}^{min} \tau_h^{min})^2}{(m+1)(\gamma_{\ell}^{min} \tau_h^{min})^2 - (\gamma_{\ell}^{max} \tau_h^{max})^2}\right]$$

Using the bound on the sensitivity $\boldsymbol{f}$ computed in (15), we obtain the desired $\epsilon_{PDFA}$, upper bound of the Rényi divergence. A more detailed proof is presented in the Appendix. $\square$

**Remark 3.1.** This bound is not tight since it assumes that all the activations reach their worst cases in all the layers for upper bounding. However, obtaining a tighter bound would be very challenging since the values of the covariance matrices depend on the output of the neurons of the Neural network, which are data and architecture dependent.

We believe tighter bounds could be obtained in much simpler cases. First we can notice that having equal covariance matrices $\boldsymbol{\Sigma}$ and $\boldsymbol{\Sigma}'$ would cancel the logarithm term. If additionally we assume that all the activations saturate to their clipping values, then we would retrieve the formula of $\epsilon$ in [19].

Owing to mini-batch training, we believe the privacy parameter could be further improve by considering subsampling mechanism [30] and its properties. However, this would require a novel theoretical framework adapted to our case and we leave it for future work.

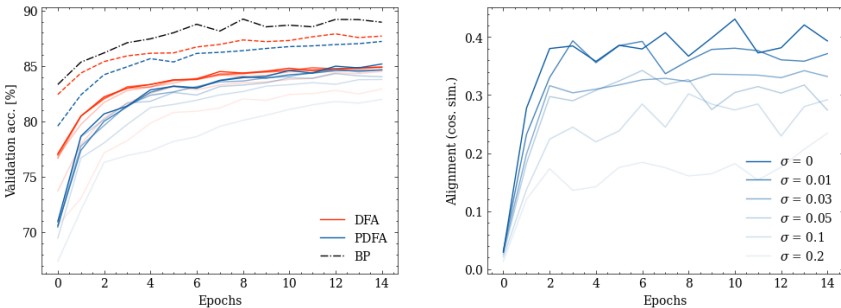

Figure 2: **Photonic training on FashionMNIST.** Left: BP, DFA, and photonic DFA (PDFA) training runs for various degrees of privacy. Dashed runs (**--**) are non-private. Increasingly transparent runs have increased noise, see Table 1 for details. PDFA is always very close to DFA performance, and both are robust to noise. Right: gradient alignment (cosine similarity between PDFA and BP gradients) for the second layer of the network, at varying degrees of noise. Increasing noise degrades alignment, but alignment values remain high enough to support learning.

## 4    Experimental results

In this section, we demonstrate that photonic training is robust to Gaussian mechanism, i.e. adding noise as in Eq. 8, delivering good end-task performance even under strong privacy constraints. As detailed by our theoretical analysis, we focus on two specific mechanisms:

- **Clipping and offsetting the neurons** with $\frac{\tau_h^{max}}{\sqrt{n_\ell}}$ and $\frac{\tau_h^{min}}{\sqrt{n_\ell}}$ to enforce $\tau_h^{min} \leq \|\boldsymbol{h}^l\|_2 \leq \tau_h^{max}$, as explained in section 3.1;
- **Adding noise g to Be**, according to the clipping of $\mathbf{Be}$ with $\tau_f$ ($\|\mathbf{Be}\|_2 \leq \tau_f$) and the scaling of $\mathbf{g}$ with $\sigma$ ($\mathbf{g} \sim \mathcal{N}(0, \sigma^2\mathbf{I})$).

To make our results easy to interpret, we fix $\tau_f = 1$, such that $\sigma = 0.1$ implies $\|\mathbf{g}\|_2 \simeq \|\mathbf{Be}\|_2$. At $\sigma = 0.01$, this means that the noise is roughly 10% of the norm of the DFA learning signal. This is in line with differentially-private setup using backpropagation, and is in fact more demanding as our experimental setup makes it such that this is a lower bound on the noise.

**Photonic DFA.**    We perform the random projection $\mathbf{Be}$ at the core of the DFA training step using the OPU, a photonic co-processor. As the inputs of the OPU are binary, we ternarize the error – although binarized DFA, known as Direct Random Target Propagation [11], is possible, its performance is inferior. Ternarization is performed using a tunable treshold $t$, such that values smaller than $-t$ are set to -1, values larger than $t$ are set to 1, and all in between values are set to 0. We then project the positive part $\mathbf{e}_+$ of this vector using the OPU, obtaining $\mathbf{Be}_+$, and then the negative part $\mathbf{e}_-$, obtaining $\mathbf{Be}_-$. Finally, we substract the two to obtain the projection of the ternarized error, $\mathbf{B}(\mathbf{e}_+ - \mathbf{e}_-)$. This is in line with the setup proposed in [18]. We refer thereafter to DFA performed on a ternarized error on a GPU as ternarized DFA (TDFA), and to DFA performed optically with a ternarized error as photonic DFA (PDFA).

**Setting.**    We run our simulations on cloud servers with a single NVIDIA V100 GPU and an OPU, for a total estimate of 75 GPU-hours. We perform our experiments on FashionMNIST dataset [32], reserving 10% of the data as validation, and reporting test accuracy on a held-out set. We use a fully-connected network, with two hidden layers of size 512, with tanh activation. Optimization is done over 15 epochs with SGD, using a batch size of 256, learning rate of 0.01 and 0.9 momentum. For TDFA, we use a threshold of 0.15. Despite the fundamentally different hardware platform, no specific hyperparameter tuning is necessary for the photonic training: this demonstrates the reliability and robustness of our approach.

**BP baseline.**    We also apply our DP mechanism to a network trained with backpropagation. The clipping and offsetting of the activations' neurons is unchanged, but we adapt the noise element. We

Table 1: **Test accuracy on FashionMNIST with our DP mechanism.** We find our approach to be robust to increasing DP noise $\sigma$. In particular, photonic DFA results (PDFA) are always within 1% of the corresponding DFA run.

| $\sigma$
$\tau_f$ | non-private | 0 | 0.01 | 0.03 | 0.05
1 | 0.1 | 0.2 |
|------|-------------|-------|-------|-------|-------|-------|-------|
| **BP** | 88.33 | 75.22 | 70.71 | 71.47 | 71.27 | 70.28 | 66.78 |
| **DFA** | 86.80 | 84.20 | 84.04 | 84.15 | 83.70 | 83.06 | 81.66 |
| **TDFA** | 86.63 | 84.20 | 84.38 | 84.04 | 83.94 | 82.98 | 80.80 |
| **PDFA** | 85.85 | 84.00 | 83.79 | 83.69 | 83.36 | 82.63 | 80.94 |

apply the noise on the derivative of the loss once at the top of the network. We also lower the learning rate to $10^{-4}$ to stabilize runs.

**Results.** We fix $\tau_h^{\max} = 1$ for all experiments, and consider a range of $\sigma$ corresponding to noise between 0-200% of the DFA training signal **Be**. We also compare to a non-private, vanilla baseline. Results are reported in Table 1 and Figure 2.

We find our DFA-based approach to be remarkably robust to the addition of noise, providing Differential Privacy, with a test accuracy hit contained within 3% of baseline for up to $\sigma = 0.05$ (i.e. noise 50% as large as the training signal). Most of the performance hit can actually be attributed to the aggressive activation clipping, with noise having a limited effect. In comparison, BP is far more sensitive to activation clipping and to our noise mechanism. However, our method was devised for DFA and not BP, explaining the under-performance of BP. Finally, photonic training achieves good test accuracy, always within 1% of the corresponding DFA run. This demonstrates the validity of our approach, on a real photonic co-processor. We note that, usually, demonstrations of neural networks with beyond silicon hardware are mostly limited to simulations [14, 12], or that these demonstrations come with a significant end-task performance penalty [34, 31].

**Additional results** with similar conclusions on MNIST and CIFAR-10 are presented in the Appendix.

## 5 Conclusion and Outlooks

We have investigated how the Gaussian measurement noise that goes with the use of the photonic chips known as Optical Processor Units, can be taken advantage of to ensure a Differentially Private Direct Feedback Alignment training algorithm for deep architectures. We theoretically establish the features of the so-obtained *Photonic Differential Privacy* and we feature these theoretical findings with compelling empirical results showing how adding noise does not decreases the performance significantly.

At an age where both privacy-preserving training algorithms and energy-aware machine learning procedures are a must, our contribution addresses both points through our photonic differential privacy framework. As such we believe the holistic machine learning contribution we bring will mostly bring positive impacts by reducing energy consumption when learning from large-scale datasets and by keeping those datasets private. On the negative impact side, our DP approach is heavily based on clipping, which is well-known to have negative effects on underrepresented classes and groups [5, 13] in a machine learning model.

We plan to extend the present work in two ways. First, we would like to refine the theoretical analysis and exhibit privacy properties that are more in line with the observed privacy; this would give us theoretical grounds to help us set parameters such as the clipping thresholds or the noise modulation. Second, we want to expand our training scenario and address wider ranges of applications such as recommendation, federated learning, natural language processing. We also plan to spend efforts so as to mitigate the effect of clipping on the fairness of our model [33].

## Acknowledgments

R.O. acknowledges funding from the Région Ile-de-France. The authors would like to thanks Kilian Muller and Gustave Pariente for fruitful discussions regarding the OPU. This work was granted access to the HPC/AI resources of IDRIS under the allocation 2021-A0101012429 made by GENCI. Specifically, we thank the team of the Jean Zay supercomputer for their support.

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
