# OpenReview forum: "Photonic Differential Privacy with Direct Feedback Alignment"
_NeurIPS.cc/2021/Conference — NeurIPS 2021 Poster_

### Official Review · Reviewer_bXri · 2021-07-10

**Rating:** 6
**Confidence:** 3

**Summary:**

The manuscript entitled “Photonic Differential Privacy with Direct Feedback Alignment” extends their previous work - Differential Privacy with Direct Feedback Alignment. In this work, they leverage the intrinsic noise (which might be controlled by temperature) of optical random projections to build a differentially private DFA mechanism, making Optical Processing Units (OPUs) a solution of choice to provide a private-by-design training. This work is motivated by a photonic implementation that naturally induces noise that we exploit for differential privacy. The proposed method is demonstrated by simulations showing that adding noise does not decrease the performance significantly. In other words, this approach is robust to increasing DP noise. This paper also provides its theoretical analysis.

**Limitations And Societal Impact:**

Yes, they are addressed adequately.

**Main Review:**

This paper has some significance in terms of making Photonic Differential Privacy with Direct Feedback Alignment suitable for protecting privacy in training by nature. I have a few comments and hope them can be addressed before its publication:

1.	On Line 14 Page 4, what is the relation between χ and χ_n?

2.	In definition 2.2 on Page 4, it says A : D ->Im A means Im A is the image of D through A, and O ∈ Im A. How can A(D) which is Im A belong to O, i.e. A(D) ∈ O?

3.	Neither Figure 1 nor Figure 2 are discussed in any section. The authors should discuss them in the corresponding sections.

4.	In Figure 2, the authors use the degree of transparency of the lines to indicate the noise level. However, with more transparency, it becomes quite hard to distinguish the colors of the lines (e.g blue vs black). It will become even worse if we print the paper out. Figure 2 should be revised with other line styles.

5.	In Table 1 Page 9, what is the difference between non-private and sigma=0?

6.	On Line 215 Page 8, it should be ‘improved’ instead of ‘improve’. On Line 31-32 Page 8, the sentence should be rephrased to avoid grammar mistakes.

In summary, I recommend its publication in NeurIPS after those commends are addressed.


**Time Spent Reviewing:**

4

---

> ### Author Response · Authors · 2021-08-11
> **Answer to Reviewer bXri**
>
> We thank the reviewer for the time taken to review our paper and for their useful feedback. We will make sure to implement in the camera-ready version all comments regarding clarity, notation, and formatting (3-4, and 6). We address below specific questions & points raised in the review.
>
> > The manuscript entitled “Photonic Differential Privacy with Direct Feedback Alignment” extends their previous work - Differential Privacy with Direct Feedback Alignment.
>
> **We would like to clarify that this work is not an extension of the DPDFA paper**. The present work is an independent work which puts at its core the use of photonic devices, which enables differential privacy "by design" and which differs in its procedure from the work mentioned.
>
> > 1. On Line 14 Page 4, what is the relation between χ and χ_n?
>
> $X$ is the domain of a datapoint ($R^d$ usually) and $X\_n$ is the domain of $n$’th datapoint so **they correspond to the same quantity**.
>
> > 2. In definition 2.2 on Page 4, it says A : D ->Im A means Im A is the image of D through A, and O ∈ Im A. How can A(D) which is Im A belong to O, i.e. A(D) ∈ O?
>
> **This is a typo, thank you for carefully checking**. Indeed, this should be A:D $->$ Range(A) instead of A:D $->$ Im A.
>
> > 5. In Table 1 Page 9, what is the difference between non-private and sigma=0?
>
> Non-private corresponds to a completely "vanilla" setting, with no differential privacy mechanisms enabled. **The $\sigma=0$ case is provided as an "ablation" of the DP mechanism**: in this case, there is no extra noise, but the clipping operation is still performed. This shows that the clipping is responsible for most of the drop in accuracy, and that DFA is more resilient to it than BP.
>
> We hope that these clarifications are sufficient for the reviewer to recommend the publication of our paper at NeurIPS.

---

> > ### Comment · Reviewer_bXri · 2021-08-21
> > **Thanks for the clarification**
> >
> > I keep my rating

---

### Official Review · Reviewer_BUbz · 2021-07-16

**Rating:** 5
**Confidence:** 5

**Summary:**

The paper proposes a hardware-based approach to Differential Privacy (DP) using OPU to perform optical random projections for a differentially private DFA training algorithm, leveraging noise intrinsic to the hardware to achieve privacy-by-design. The proposed approach was analyzed mathematically in detail. The proposed methods were analyzed on the vanilla Fashion MNIST dataset.

The paper is very well written in general. The proposed methods/approach are very interesting, and mathematical results are solid.


**Limitations And Societal Impact:**

Please see the above comments.

**Main Review:**

One of the major issues with the paper is limited experimental analysis of the proposed methods in comparison with baseline DFA and state-of-the-art DP methods on different benchmark datasets. Although mathematical results are interesting, more detailed additional experimental results are needed to examine and justify its merit compared to the state-of-the-art in different tasks.
Additional comments and suggestions are provided as follows:

1.	Please definite the notation and terms as they are used. For instance, please either define the notation used in (1) or just refer to (4).

2.	What do you mean by “train deep architectures”? I suppose that you refer training deep learning models whose functional structures are identified by the corresponding network architectures, instead of “designing network architectures using data”.

3.	Could you please provide additional analyses using more complicated datasets such as Cifar-10/100 or variations of ImageNet,  and networks, such as ResNets? Indeed, how would random projection of the error will be applied to networks with bottlenecks, skip connections and dense connections?

4.	Please check the consistency of the notation and fix it in the text. For instance, please update the notation used for distribution D, and data spaces/sets X and Y according to that used in Section 2.3, vice versa.

5.	In Definition 2.1, please define the record.

6.	 In Theorem 1, please define the mechanism M_i, more precisely.

7.	Please define | \cdot |

8.	Please define || || used at line 148, page 5.

9.	Page 5, line 162: Section 3.1

10.	Alg. 1, line 15: g_1^l,…,g_m^l

11.	Please fix use of gaussian or Gaussian.

12.	Please capitalize section names.

13.	 In Proposition 13, what is the difference between .n_l\alpha and n_l.\alpha?

14.	Please provide a complexity analysis of the proposed method in comparison with non-private and baseline DFA.

After Rebuttal: Thank you for your response to my comments. I checked the comments of other reviewers and the response to their comments as well. The authors answered most of the questions in the rebuttal. Therefore, I increase my grade, while I consider that the paper should be improved for a clear acceptance, especially with additional experimental results.

**Time Spent Reviewing:**

4

---

> ### Author Response · Authors · 2021-08-11
> **Answer to reviewer BUbz**
>
> We thank the reviewer for the time taken to review our paper and for their useful feedback, which helps strengthen our paper. We will make sure to implement in the camera-ready version all comments regarding clarity, notation, and formatting (1-2, 4-12). We answer below to point 3 & 14 specifically, as well as more general comments by the reviewer.
>
> > One of the major issues with the paper is limited experimental analysis of the proposed methods in comparison with baseline DFA and state-of-the-art DP methods on different benchmark datasets.
>
> **The primary focus of our paper is to demonstrate the possibility of using specialised hardware in combination with an alternative to backpropagation to implement differential privacy.** A key feature of our approach is that the DP mechanism comes "built-in" with the hardware, leveraging its physical properties. This is unique, and in line with recent interest around work co-designing ML algorithms and hardware together [1]. We provide theoretical backing for our approach, as well as an empirical validation.
>
> In our empirical demonstration, we focused on showing that our algorithmic contributions & approach worked in practice, and to assess the impact of differentially private learning for photonic DFA in a controlled setting. We compared our method directly to "vanilla" DFA and backpropagation, both with our DP scheme. **We want to highlight the fact that so-called optical or photonic training is still very novel, and rarely demonstrated on actual hardware as it is here**. Applying other DP approaches to DFA and photonic DFA would require significant rework of either the approach or DFA, and we feel this is better left to a future contribution.
>
> > Could  you  please  provide  additional  analyses  using  more  complicated  datasets  such  as  Cifar-10/100 or variations of ImageNet, and networks, such as ResNets?
>
> **We provide below results on MNIST** (requested by reviewer `MLtY`) obtained with the same code, procedure, and hyper-parameters as for FashionMNIST. These results are in line with Table 1 of our paper, with photonics results always close to ternarized ones. We need that this "default" choice of hyper-parameters on MNIST results in ternarized DFA outperforming vanilla DFA. (We only lightly tune hyperparameters on BP, to demonstrate that our approach does not require any specific expensive fine-tuning search.)
>
> |      	| non-private 	| $\sigma$ 	|       	|       	|
> |:----:	|:-----------:	|----------	|:-----:	|:-----:	|
> |      	|             	|   0.01   	|  0.05 	|  0.1  	|
> | BP   	| 97,94       	| 62,63    	| 58,42 	| 48,33 	|
> | DFA  	| 96,36       	| 92,99    	| 92,68 	| 92,45 	|
> | TDFA 	| 97,09       	| 93,67    	| 93,57 	| 93,28 	|
> | ODFA 	| 96,95       	| 93,60    	| 93,57 	| 93,12 	|
>
> These results will be added to our supplementary material.
>
> **We are also in the process of obtaining results on CIFAR-10** through fine-tuning of the classifier with photonic DFA; we will comment back in a few days with these results. This fine-tuning/transfer learning approach is the path to scaling our approach to "real-world" datasets such as ImageNet.
>
> > Indeed, how would random projection of the error will be applied to networks with bottlenecks, skip connections and dense connections?
>
> A key point we will highlight in our updated version is that DFA does not work with convolutions [2, 3]. This prevents its use in a ResNet; however **it can still be used for fine-tuning the classifier in a differentially private way** (which we will explore for the CIFAR-10 results mentioned above).
>
> The question of adapting DFA to various architecture is beyond the scope of our paper, as complex architecture-specific considerations have to be taken into account. We point the reviewer to [3] for work along this line, demonstrating and adapting DFA in a number of modern deep learning architectures.
>
> To answer specifically the question on ResNets, one of the key feature of DFA is that it unlocks the backward pass, enabling each layer to be processed in parallel after the forward pass. **This wouldn't be compatible with residual connections, which would remove that parallelism in the backward pass**. [3] includes Transformers, which feature a very rough equivalent of skip connections (through a residual path). In that case, the skip connection was made assymmetric, allowing information to flow in the forward between layers, but not in the backward.
>
> > Please provide a complexity analysis of the proposed method in comparison with non-private and baseline DFA.
>
> We have performed the following numerical analysis and will include it in the paper before its publication.
>
> Let's focus on layers $\ell$ and $\ell + 1$. For BP, matrix-vector multiplication between the layer weights $W^{\ell +1 }$ and $\delta z^{\ell+1}$ costs O($n_\ell n_{\ell +1}$), the Hadamard product is of linear cost so not included in the total complexity cost, and the outer product with $h^{\ell -1}$ is of cost O($n_\ell n_{\ell -1}$). This yields a total cost of O($n_\ell( n_{\ell +1} + n_{\ell -1})$). This is the same cost for DFA since $B^{\ell +1}$ is of the size of $W^{\ell +1}$. However, performing matrix-vector multiplication with the OPU is of cost O(1) [5] which implies that $B^{\ell +1} e$ is of cost O(1). Therefore, after taking into account the cost of the outer product, the total cost of Photonic DFA is O($n_\ell n_{\ell -1}$).
>
> **This further motivates the use of the OPU to perform DFA, as photonic DFA is thus of lower complexity than both BP and DFA**.
>
> We hope that these clarifications and additional details are sufficient for the reviewer to recommend the publication of our paper at NeurIPS.
>
> ### References
> [1]: Hooker. "The Hardware Lottery". arXiv pre-print.
>
> [2]: Launay, et al. "Principled Training of Neural Networks with Direct Feedback Alignment." arXiv pre-print.
>
> [3]: Refinetti, et al. "Align, then memorise: the dynamics of learning with feedback alignment." Proceedings of the 38th International Conference on Machine Learning (ICML 2021).
>
> [4]: Launay, et al. "Direct Feedback Alignment Scales to Modern Deep Learning Tasks and Architectures." 34th Conference on Neural Information Processing Systems (NeurIPS 2020).
>
> [5]: Ohana, et al. "Kernel computations from large-scale random features obtained by optical processing units." IEEE International Conference on Acoustics, Speech and Signal Processing (ICASSP 2020).

---

> > ### Comment · Reviewer_BUbz · 2021-08-25
> > **Reply to rebuttal**
> >
> > Thank you for your response to my comments. I checked the comments of other reviewers and the response to their comments as well. The authors answered most of the questions in the rebuttal. Therefore, I increase my grade, while I consider that the paper should be improved for a clear acceptance, especially with additional experimental results.

---

> ### Author Response · Authors · 2021-08-24
> **Final answer to reviewer BUbz: results on CIFAR-10**
>
> We thank the reviewer for their patience regarding the results on CIFAR-10. We provide this answer as a complement to our previous answer, which addressed the reviewer's other points.
>
> We thank again the reviewer for their time and recommendations, which have helped strengthen our paper. We wish that our responses have addressed your concerns, and turns your assessment to the positive side.
>
>
> > Could you please provide additional analyses using more complicated datasets such as Cifar-10/100 or variations of ImageNet, and networks, such as ResNets?
>
> Regarding the CIFAR-10 experiment, we chose to use a pre-trained network on ImageNet and extract its trained convolutional layers. Since these convolutions can be seen as feature extractors of the images and are not re-trained, they do not need to be taken into account into the Differential Privacy mechanism. We fine-tune only the fully-connected layers of the classifier using our Photonic DFA+DP mechanism. This is inline with the classic setup in [1].
>
> We choose to focus our CIFAR-10 experiment on demonstrating the scalability of our scheme. We do not seek to achieve state-of-the-art performance or to exhaustively explore the dynamics/impact of different differentially private configuration (as we did with MNIST), but simply to show our scheme can scale to such harder tasks.
>
> We used a VGG16 network pre-trained on ImageNet. We leave the convolutions untouched, and fine-tune the classifier layers (25088 --> 4096 --> 4096 --> 10) with differentially private photonic training. We do not use any data augmentation, and simply resize the CIFAR-10 images to 224x224. We fine-tune for 15 epochs, using SGD with learning rate 5e-3, momentum 0.9, and batch size 256. Hyperparameters are kept identical across all methods and hardware. We obtain results both in a vanilla (no DP) setting as a comparison baseline, and in a differentially private setting with $\sigma=0.05$.
>
> **Vanilla (no differential privacy):** 83.17% (backpropagation), 81.34% (DFA), 83.36% (ternarized DFA).
> **DP ($\sigma=0.05$):** 60.45% (BP), 79.68% (DFA), 79.33% (TDFA), 78.64% (photonic DFA).
>
> We note that this result shows good scalability, with performance in line with our MNIST/FashionMNIST results. Our DP scheme does not result in too large of a performance loss with DFA/TDFA/PDFA, and our photonic results are within 1% of results simulated on a GPU.
>
> We will be adding a section in the camera-ready version of the paper paper regarding scaling-up with fine-tuning pre-trained networks. We will include the results above in this section.
>
> We thank again the reviewer for suggesting these experiments, as we feel they make the narrative of our paper more compelling.
>
> [1] Abadi, M., Chu, A., Goodfellow, I., McMahan, H.B., Mironov, I., Talwar, K. and Zhang, L., 2016, October. Deep learning with differential privacy. In Proceedings of the 2016 ACM SIGSAC conference on computer and communications security (pp. 308-318).

---

### Official Review · Reviewer_MLtY · 2021-07-18

**Rating:** 7
**Confidence:** 4

**Summary:**

This work proposes leveraging the intrinsic noise induced by optical processors performing matrix multiplication to provide differential privacy to neural networks trained with the direct feedback alignment algorithm. The authors demonstrate both theoretically and empirically the viability of this approach by training a neural network on an optical processing unit with direct feedback alignment and providing multiple levels of differential privacy noise.

**Limitations And Societal Impact:**

Some suggestions for improvement:

-Further detail on properties of the optical processing unit that make it particularly well suited for direct feedback alignment (the intrinsic noise justifies the differential privacy)

-Some experiments beyond the FashionMNIST dataset would also be helpful to demonstrate whether the performance differences widen or narrow with dataset difficulty (possibly CIFAR10 and MNIST).

-In the context of real-world privacy and what differential privacy implies, it would be helpful to provide a specific example use case. The formulations (Definition 2.2) imply that private information can still be present in the network and only the notion of plausible deniability of what data the model was trained on. This may be beyond the scope of this particular paper but would help make a more complete standalone work.

**Main Review:**

This work demonstrates what may have originally been considered a weakness of analog hardware to be a benefit in the context of data privacy. Furthermore, they demonstrate the real-world applicability of this approach by implementing the proposed learning procedure on an actual optical processing device. But details about the operations performed by the optical processing unit are not reported and what features about this device make it particularly well suited for the direct feedback alignment compared to other neural network learning procedures (most notably the backpropagation algorithm). There is some mention about the inherent parallelizability of the layerwise training with direct feedback alignment as compared to backpropagation, but this is generally a benefit to most neural network hardware solutions including CPUs/GPUs. Therefore, further justification about the chosen learning algorithm with the optical processing unit hardware should be expanded upon. It's also a little unintuitive why the optical processing is only used in the matrix multiplication of the DFA backward pass while a conventional GPU (nvidia v100) is used for the forward pass. Does this not limit the advantages of the proposed procedure since the conventional GPU creates a lower bound in training time?

Overall, the work is an exciting demonstration of the viability of the direct feedback alignment algorithm to a novel hardware domain. But this excitement is somewhat muted by the lack of detail of why combining these two things makes sense and more detail in this respect would further improve this work.

**Time Spent Reviewing:**

3

---

> ### Author Response · Authors · 2021-08-11
> **Answer to reviewer MLtY**
>
> We thank the reviewer for the time taken to review our paper and for their useful feedback. We address the suggestions made below.
>
> > details  about  the  operations  performed  by  the  optical  processing  unit  are  not  reported  and  what features about this device make it particularly well suited for the direct feedback alignment compared to other neural network learning procedures.
>
> The operation implemented by **the OPU is described in section 2.2, more specifically equation 5**: it is the projection of an input by a fixed random projection with Gaussian distribution. This turns out to be precisely the operation central to DFA, i.e. a random projection of the error vector.
>
> This random projection in DFA may be computationally expensive in large networks, especially in terms of memory. This has prevented past papers to scale DFA to ImageNet [1]. **Using the OPU offloads this computation from the GPU, making it come at no memory cost**. Furthermore, the OPU uses only ~30W, compared to the >250W required for modern GPUs, making it significantly more energy efficient for such operations.
>
> Finally, given the comment raised by Reviewer `BUbz`, we have also derived the computational complexity of the weight updates and showed that **photonic DFA has a lower complexity with respect to BP and baseline DFA** (thanks to the complexity of the RP on the OPU being independent of its size [2]).
>
> We will add to the camera-ready version a few lines on the above motivation, clarifying why DFA and the OPU are a natural fit.
>
> > It's also a little unintuitive why the optical processing is only used in the matrix multiplication of the DFA backward pass while a conventional GPU (nvidia v100) is used for the forward pass. Does this not limit the advantages of the proposed procedure since the conventional GPU creates a lower bound in training time?
>
> The OPU can only be used in the backward pass as **it only implements random matrix multiplication**, which is not featured in the forward pass. For large dimensions (~$10^5 \times 10^5$), the OPU is faster than a GPU [2] at performing the random projection; and indeed, the GPU will become the bottleneck for training. However, our procedure retains a number of advantages :
> * As outlined above, the random matrix $B$ used in DFA doesn't have to be stored on the GPU, saving memory;
> * The privacy-inducing noise comes "built-in" with the OPU doing the backward pass.
>
> From a more general perspective, **the motivation of our approach is to propose a hybrid approach combining both the strengths of the OPU and GPU**.
>
> > Some experiments beyond the FashionMNIST dataset would also be helpful to demonstrate whether the performance differences widen or narrow with dataset difficulty (possibly CIFAR10 and MNIST).
>
> **We provide below results on MNIST**, obtained with the same code, procedure, and hyper-parameters as for FashionMNIST. These results are in line with Table 1 of our paper, with photonics results always close to ternarized ones. We need that this "default" choice of hyper-parameters on MNIST results in ternarized DFA outperforming vanilla DFA. (We only lightly tune hyperparameters on BP, to demonstrate that our approach does not require any specific expensive fine-tuning search.)
>
> |      	| non-private 	| $\sigma$ 	|       	|       	|
> |:----:	|:-----------:	|----------	|:-----:	|:-----:	|
> |      	|             	|   0.01   	|  0.05 	|  0.1  	|
> | BP   	| 97,94       	| 62,63    	| 58,42 	| 48,33 	|
> | DFA  	| 96,36       	| 92,99    	| 92,68 	| 92,45 	|
> | TDFA 	| 97,09       	| 93,67    	| 93,57 	| 93,28 	|
> | ODFA 	| 96,95       	| 93,60    	| 93,57 	| 93,12 	|
>
> These results will be added to our supplementary material.
>
> **We are also in the process of obtaining results on CIFAR-10** through fine-tuning of the classifier with photonic DFA; we will comment back in a few days with these results. This fine-tuning approach is the path to scaling our approach to "real-world" datasets such as ImageNet.
>
> > In  the  context  of  real-world  privacy  and  what  differential  privacy  implies,  it  would  be  helpful  to provide  a  specific  example  use  case.  The  formulations  (Definition  2.2)  imply  that  private  information  can still be present in the network and only the notion of plausible deniability of what data the model was trained on.  This may be beyond the scope of this particular paper but would help make a more complete standalone work
>
> Differential Privacy (DP) is indeed not a bulletproof standalone solution for real world privacy applications. The relaxed ($\epsilon,\delta$)-differential privacy definition can only be viewed as a model of plausible deniability, due to its probabilistic nature. In our approach, we show how to leverage the intrinsic noise of optical random projections to build a differentially private DFA mechanism, but our solution also suffers from similar limitations and caveats to the standard DP mechanisms, e.g., a Gaussian Mechanism with ($\epsilon,\delta$) parameters. This implies that most of the time the information contained in our model cannot be used to distinguished a given private input record, but a few times it can; more specifically, any ($\epsilon,\delta$)-DP mechanism has a probability  $\delta$ of leaking some amount of private information. In practical settings, a critical point to focus on is in trading off an effective balance between the amount of privacy we require for a given applications and the utility of algorithms.
>
> We will add a few lines discussing this limitation, as well as references to works using differential privacy in real-world scenario (e.g. transfer learning, where a model learned on public data has to be adapted to privacy-sensitive data).
>
> ### References
>
> [1]: Bartunov, et al. "Assessing the Scalability of Biologically-Motivated Deep Learning Algorithms and Architectures." 32nd Conference on Neural Information Processing Systems (NeurIPS 2018).
>
> [2]: Ohana, et al. "Kernel computations from large-scale random features obtained by optical processing units." IEEE International Conference on Acoustics, Speech and Signal Processing (ICASSP 2020).

---

> > ### Comment · Reviewer_MLtY · 2021-08-20
> > **Reply**
> >
> > Thank you for the responses. Just some followup questions:
> >
> > > it is the projection of an input by a fixed random projection with Gaussian distribution. This turns out to be precisely the operation central to DFA, i.e. a random projection of the error vector.
> >
> > I was mainly asking what physical principles of this OPU enables it to be an (exact?) projection with a gaussian distribution. If there were any assumptions made about the physical model to reach this conclusion.
> >
> > > We are also in the process of obtaining results on CIFAR-10 through fine-tuning of the classifier with photonic DFA; we will comment back in a few days with these results. This fine-tuning approach is the path to scaling our approach to "real-world" datasets such as ImageNet
> >
> > Have the results come back yet?

---

> > > ### Author Response · Authors · 2021-08-24
> > > **OPU distribution clarification & CIFAR-10 results**
> > >
> > > We thank the reviewer for their patience and interest in our paper.
> > >
> > > > I was mainly asking what physical principles of this OPU enables it to be an (exact?) projection with a gaussian distribution. If there were any assumptions made about the physical model to reach this conclusion.
> > >
> > > We refer the reviewer to [1] and [2], which provide an explanation of why the transmission matrix is naturally a Gaussian distribution (looking at the eigenvalue distribution is one of the tools). Note that not every diffusive medium has this property, it needs to be a multiple scattering medium and we refer to [1] for more precise experimental details about the diffusive medium.
> > >
> > > > Have the results come back yet?
> > >
> > > Regarding the CIFAR-10 experiment, we chose to use a pre-trained network on ImageNet and extract its trained convolutional layers. Since these convolutions can be seen as feature extractors of the images and are not re-trained, they do not need to be taken into account into the Differential Privacy mechanism. We fine-tune only the fully-connected layers of the classifier using our Photonic DFA+DP mechanism. This is inline with the classic setup in [3].
> > >
> > > We choose to focus our CIFAR-10 experiment on demonstrating the scalability of our scheme. We do not seek to achieve state-of-the-art performance or to exhaustively explore the dynamics/impact of different differentially private configuration (as we did with MNIST), but simply to show our scheme can scale to such harder tasks.
> > >
> > > We used a VGG16 network pre-trained on ImageNet. We leave the convolutions untouched, and fine-tune the classifier layers (25088 --> 4096 --> 4096 --> 10) with differentially private photonic training. We do not use any data augmentation, and simply resize the CIFAR-10 images to 224x224. We fine-tune for 15 epochs, using SGD with learning rate 5e-3, momentum 0.9, and batch size 256. Hyperparameters are kept identical across all methods and hardware. We obtain results both in a vanilla (no DP) setting as a comparison baseline, and in a differentially private setting with $\sigma=0.05$.
> > >
> > > **Vanilla (no differential privacy):** 83.17% (backpropagation), 81.34% (DFA), 83.36% (ternarized DFA).
> > > **DP ($\sigma=0.05$):** 60.45% (BP), 79.68% (DFA), 79.33% (TDFA), 78.64% (photonic DFA).
> > >
> > > We note that this result shows good scalability, with performance in line with our MNIST/FashionMNIST results. Our DP scheme does not result in too large of a performance loss with DFA/TDFA/PDFA, and our photonic results are within 1% of results simulated on a GPU.
> > >
> > > We will be adding a section in the camera-ready version of the paper paper regarding scaling-up with fine-tuning pre-trained networks. We will include the results above in this section.
> > >
> > > We thank again the reviewer for suggesting these experiments, as we feel they make the narrative of our paper more compelling.
> > >
> > > [1] Popoff, S. M. et al. Measuring the transmission matrix in optics: an approach to the study and control of light propagation in disordered media. Phys. Rev. Lett. 104, 100601 (2010)
> > > [2] Popoff, S. M.e t al. Controlling light through optical disordered media: transmission matrix approach. New J. Phys. 13, 123021 (2011).
> > > [3] Abadi, M., Chu, A., Goodfellow, I., McMahan, H.B., Mironov, I., Talwar, K. and Zhang, L., 2016, October. Deep learning with differential privacy. In Proceedings of the 2016 ACM SIGSAC conference on computer and communications security (pp. 308-318).

---

### Comment · Area_Chair_6Baw · 2021-08-27
**Question re. noise distribution.**

The noise added by the OPU is modeled as a Gaussian (a.k.a. normal) distribution. How is this assumption justified? Is there a physical model to support this or experimental results? A fully rigorous privacy analysis would need to account for any deviation of the actual distribution from a true Gaussian.

Lines 29-31 say "The main sources of noise in Optical Processing Units can be modeled as additive Poisson noise on the output signal, and approach a Gaussian distribution in the operating regime of the device." Can you elaborate on this?

I do not understand the physics of OPUs, but it seems to me that, if the signals are transmitted by means of light, then there is no way to transmit a negative number. This would limit how close the values can be to truly Gaussian.

I would also point out that there is a lot of interest n generating true randomness (as opposed to pseudorandom number generation). E.g. https://en.wikipedia.org/wiki/Hardware_random_number_generator How does the noise from an OPU fit into this picture?

---

> ### Author Response · Authors · 2021-08-31
> **Answer to Area Chair 6Baw**
>
> We would like to thank the Area Chair for the attention given to our paper and the interesting questions raised.
>
> > The noise added by the OPU is modeled as a Gaussian (a.k.a. normal) distribution. How is this assumption justified? Is there a physical model to support this or experimental results? A fully rigorous privacy analysis would need to account for any deviation of the actual distribution from a true Gaussian.
>
> > Lines 29-31 say "The main sources of noise in Optical Processing Units can be modeled as additive Poisson noise on the output signal, and approach a Gaussian distribution in the operating regime of the device." Can you elaborate on this?
>
> The main source of noise is the photon shot noise (coming from the fact that light arrive in quanta of photons, with random arrival times) and is modelled as Poisson distribution (see https://en.wikipedia.org/wiki/Poisson_distribution for conventions) where $k$ (events) is the number of photons hitting the sensor. The full well of the acquisition device can contain $\sim10^4$ electrons, which translates to $k \sim 10^4$ for a fully bright image. For a good Signal-to-noise Ratio, we are typically at $k \sim 10^3$, so $\lambda$ = a few hundreds. In this regime, the Poisson distribution is approximated by a Gaussian by performing the continuity correction $\mathbb{P}(X≤x) = \mathbb{P}(Y≤x+\frac12)$ ($X$ is Poisson, $Y$ is Gaussian). References are given in [1],[2].
>
>
> > I do not understand the physics of OPUs, but it seems to me that, if the signals are transmitted by means of light, then there is no way to transmit a negative number. This would limit how close the values can be to truly Gaussian.
>
> It is easy to post-process such that the output values can be positive or negative while still maintaining additive Gaussian noise. For instance, imagine two pixels (which are positive) $p_1= x + \mathcal{N}(\mu, \sigma^2)$  and $p_2= y + \mathcal{N}(\mu, \sigma^2)$ , one can do differential detection (subtract half of the values in the output from the other half), and obtain $p_1- p_2 = x - y + \mathcal{N}(2\mu, 2\sigma^2)$ which can be negative.
>
>
> > I would also point out that there is a lot of interest in generating true randomness (as opposed to pseudorandom number generation). E.g. https://en.wikipedia.org/wiki/Hardware_random_number_generator How does the noise from an OPU fit into this picture?
>
> Photon shot noise (the main source of noise in the system) is a source of true randomness due to the uncertainty principle in quantum mechanics. It is however difficult to collect it reliably for the typical use cases of true RNGs. Cf. see the subsection “Quantum random properties” in the Wikipedia page.
>
> [1] CMV2000: 2MP global shutter CMOS image sensor for machine vision. url: https://ams.com/cmv2000.
> [2] Mikhail Konnik and James Welsh. “High-level numerical simulations of noise in CCD and CMOS photosensors: review and tutorial”. In: arXiv preprint arXiv:1412.4031 (2014).

---

> > ### Comment · Area_Chair_6Baw · 2021-09-01
> > **Thanks for the clarifications.**
> >
> > I hope you will add some of these comments to the paper, to the extent that the page limit permits. The fact that OPUs generate true randomness is a selling point, as most implementations of differentially private deep learning currently use low-quality pseudorandom noise.
> >
> > Re. using differences to express negative numbers: It is worth noting that the difference of two Poisson random variables is known as a Skellam distribution and Skellam noise has been analyzed in the context of differential privacy. E.g. see Theorem 10 in https://arxiv.org/abs/1710.02036

---

> > > ### Author Response · Authors · 2021-09-02
> > > **Answer**
> > >
> > > > I hope you will add some of these comments to the paper, to the extent that the page limit permits. The fact that OPUs generate true randomness is a selling point, as most implementations of differentially private deep learning currently use low-quality pseudorandom noise.
> > >
> > > We would like to thank the Area Chair for these encouraging words. This is indeed a selling point we didn’t notice and that will be added to the next iteration of the paper, along with the other remarks that were made by you and the reviewers.
> > >
> > >
> > > > Re. using differences to express negative numbers: It is worth noting that the difference of two Poisson random variables is known as a Skellam distribution and Skellam noise has been analyzed in the context of differential privacy. E.g. see Theorem 10 in https://arxiv.org/abs/1710.02036
> > >
> > > This is an interesting remark. This reference will be added when the OPU part will be rephrased to shed more light on it.

---

> > ### Comment · Area_Chair_6Baw · 2021-09-01
> > **Privacy parameters for experimental results**
> >
> > The experimental results state the noise and clipping parameters ($\sigma$ and $\tau$), but not the privacy parameters ($\varepsilon$ and $\delta$ or $\alpha$) that these correspond to. Would you be able to compute these?

---

> > > ### Author Response · Authors · 2021-09-02
> > > **Answer**
> > >
> > > > The experimental results state the noise and clipping parameters ($\sigma$ and $\tau$), but not the privacy parameters ( $\varepsilon$ and $\delta$ or $\alpha$) that these correspond to. Would you be able to compute these?
> > >
> > > As stated in Remark 3.1 line 208, this bound was computed in the worst case scenario, when all the activations are saturated, so it is not tight. Computing an $\varepsilon$ in this case would lead to a quantity that doesn’t make physical sense. On the other hand, we chose to increase the variance $\sigma^2$, which would correspond to decreasing the Signal-to-noise Ratio of the gradient update. To us, this is more meaningful and directly linked to an understandable quantity. If one wishes to obtain the $\varepsilon$ parameter, at a first order, it can be computed using the privacy formula of https://arxiv.org/abs/2010.03701, i.e. $\varepsilon = \frac{\alpha \Delta^2}{2\sigma^2}$.
> > >
> > >
> > > Regarding $\alpha$, it plays the same role as in standard Differential Privacy, i.e. it is tuned to adjust $\varepsilon$.
> > >
> > > Regarding $\delta$, it plays the usual role, i.e. it makes the link between Renyi Differential Privacy and Differential privacy using Theorem 2.

---

> > > > ### Comment · Area_Chair_6Baw · 2021-09-02
> > > > **Thanks**
> > > >
> > > > The $\varepsilon$ values should still be reported for the experiments, even if they end up being large, as it helps compare against prior (and future) work.

---

### Decision · Program_Chairs · 2021-09-27

**Decision:**

Accept (Poster)

**Comment:**

This paper draws a connection between differentially private machine learning and noise injected by specialty analog hardware - specifically, optical processing units. This noise is normally a downside, but this submission shows that it may, in principle, be useful for privacy.

The reviewers agree that this is an interesting connection. There are some reservations about how well-developed this approach is - e.g. limited experiments, which are simulations rather than actual OPU hardware. However, the results seem acceptable as an initial proof-of-concept for this connection.